# Modelling External Magnetic Fields of Magnetite Particles: From Micro- to Macro-Scale

**Jiangang Ku [1,2], Miguel A. Valdez-Grijalva [2], Rongdong Deng [1], Weiran Zuo [1], Qidi Chen [1], Hua Lin [1] and Adrian R. Muxworthy [2,*]**

[1] School of Zijin Mine, Fuzhou University, Fuzhou 350116, China; kkcc22@163.com (J.K.); dengrongdong111@163.com (R.D.); zuoweiran@163.com (W.Z.); chenqidi596@fzu.edu.cn (Q.C.); linhua_andy@163.com (H.L.)

[2] Natural Magnetism Group, Department of Earth Science and Engineering, Imperial College London, London SW7 2AZ, UK; m.valdez-grijalva13@imperial.ac.uk

[*] Correspondence: adrian.muxworthy@imperial.ac.uk; Tel.: +44-20-7594-6442

**Abstract:** We determine the role of particle shape in the type of magnetic extraction processes used in mining. We use a micromagnetic finite element method (FEM) to analyze the effect of external magnetic fields on the magnetic structures of sub-micron magnetite particles. In non-saturating fields, the magnetite particles contain multiple possible non-uniform magnetization states. The non-uniformity was found to gradually disappear with increasing applied field strength; at 100 mT the domain structure became near uniform; at 300 mT the magnetic structure saturates and the magnetization direction aligned with the field. In magnetic separation techniques, we suggest that 100 mT is the optimal field for magnetite to maximize the magnetic field with the lowest energy transfer; larger particles, i.e., >1 μm, will likely saturate in smaller fields than this. We also examined the effect of external magnetic fields on a much larger irregular particle (L × W × H = 179.5 × 113 × 103 μm) that was too large to be examined using micromagnetics. To do this we used COMSOL. The results show the relative difference between the magnitude of magnetic flux density of the particle and that of a corresponding sphere of the same volume is <5% when the distance to the particle geometry center is more than five times the sphere radius. The ideas developed in this paper have the potential to improve magnetic mineral extraction yield.

**Keywords:** micromagnetic modelling; mineral extraction; irregular shaped particle

## 1. Introduction

To extract magnetic minerals from geological material, researchers routinely use magnetic fields to manipulate particles' movements by adjusting applied magnetic fields and manipulating inter-particle interactions [1–3]. During the past two decades, considerable progress has been made in controlling magnetic particle motion [4–6]. Most researchers use basic magnetic dipole theory to calculate the net magnetic field experienced by a given particle [7–9]; however, dipole theory does not capture all the nuances of internal non-uniform micromagnetic structures, closely packed inter-particle magnetic interactions and particle shape. Geophysicists have known this for decades, but this knowledge has not been taken up by the mineral processing community; this paper makes a first attempt to address this issue.

For distal particles, the classical magnetic dipole model calculation for the magnetic field is accurate [10,11]. However, the dipole model starts to breakdown when the inter-particle distances are of the same order of magnitude as the particles' sizes [12–14]. To model particles' internal magnetic structures and inter-particle interactions, micromagnetic algorithms are commonly used [15].

Micromagnetic analysis, whether finite element or finite difference, is far more accurate than magnetic dipole models. However, they are computationally expensive [16].

In this study, we investigate first the applied magnetic field required to saturate the magnetic structure of an irregularly shaped magnetite particle using a micromagnetic finite element method (FEM) [17]. A particle with a 'typical' geometry was chosen, and the required minimum saturating field determined. The minimum required saturating field is required for two reasons: (1) producing large magnetic fields is expensive, while the minimum field is cost efficient, and (2) if we wish particles to saturate, but still magnetically interact with each other, we require that inter-grain interaction field gradients are relatively large. Next, we considered a 'typical' macro-sized magnetite particle (62 µm volume diameter, 179.5 µm along its elongated axis) commonly seen in mineral processing engineering. Its external magnetic field structure was analysed using COMSOL. We compare the COMSOL results to that of a uniformly magnetized sphere.

## 2. Theoretical Models

### 2.1. Micromagnetism Algorithm

To determine the saturating field of an irregular micromagnetic particle, we used an existing micromagnetic FEM [17]. In this technique the magnetization, *m*, of a particle is calculated by minimizing the total free energy, given by Williams and Dunlop (Equation (1)) [18].

$$E_t = \iiint \Omega \left\{ A|\nabla \boldsymbol{m}|^2 + K_1 \left( m_x^2 m_y^2 + m_x^2 m_z^2 + m_y^2 m_z^2 \right) - M_s(\boldsymbol{H_z} \cdot \boldsymbol{m}) - \frac{1}{2}(\boldsymbol{H_d} \cdot \boldsymbol{m}) \right\} dV \qquad (1)$$

where $\iiint \Omega$ is the volume integral symbol, $A$ is the exchange stiffness constant, *m* is the magnetic moment per unit volume, $K_1$ is the first magnetocrystalline anisotropy constant, $M_s$ is the saturation magnetization, $\boldsymbol{H_z}$ is the applied magnetic field, and $\boldsymbol{H_d}$ is the demagnetizing field. For reach configuration of *m*, the gradient of the free energy $E_t$ is used to calculate a new value of *m* by a modified gradient descent method. When the change in energy is considered to be negligible, then a stable local energy minima (LEM) configuration for the magnetization is reached.

### 2.2. Numerical Calculation of the Magnetic Stray Field of Large Irregular Magnetic Structures

When the applied field is strong enough, the internal magnetization within a particle becomes uniform. We determined this field using the micromagnetic algorithm detailed in Section 2.1. By simulating the magnetic field around a 'typical' macro-sized magnetite particle (62 µm volume diameter, 179.5 µm along its elongated axis) that sits within an applied saturating field, we can study the morphology of the shape of the external magnetic field around a saturated large particle. The numerical integration of the total magnetic field at any given point was determined using COMSOL 5.3a. We also determined the external magnetic field for a sphere of equal volume.

The simulations are based on FEM (grid resolution of particle is: element size 0.1–10 mm, growth rate 1.3, curvature factor 0.2, and with the split longest side divided by four) using COMSOL 5.3a. In general form, COMSOL solves the system of partial differential equations (Equation (2)).

$$\mathbf{H} = -\nabla V_m. \qquad (2)$$

$$\nabla \cdot (\mu_0 \mathbf{H} + \mu_0 \mathbf{M}) = 0. \qquad (3)$$

$$\mathbf{B} = \mu_0(\mathbf{H} + \mathbf{M}). \qquad (4)$$

where $V_m$ is the magnetic vector potential, $\mu_0$ is the vacuum permeability, **H** and **B** represent the magnetic field, and **M** is the magnetization.

The difference between magnetic fields around the irregular particle and a sphere in the 3D coordinates cannot be easily observed. Herein, Equation (5) is used to obtain the relative difference using the magnitude of sphere magnetic flux density (MFD) as the reference value.

$$R_d(r) = \frac{norm\ B(\mathrm{r})_{particle} - norm\ B(\mathrm{r})_{sphere}}{norm\ B(\mathrm{r})_{sphere}} \times 100\% \tag{5}$$

where the $R_d$ corresponds to the relative difference, a non-dimensional number; $norm\ B(\mathrm{r})_{particle}$ and $norm\ B(\mathrm{r})_{sphere}$ correspond to the magnitude of MFD of the magnetic particle and the sphere along axis, respectively.

### 3. Results and Discussion

The micromagnetic structure as a function of applied field **H** for a micron-sized magnetite particle was modeled. The particle was 297.8 nm along its elongated axis and had an axial ratio of 1.57. It was chosen to be elongated along the hard axis (001), and the angle between the direction of applied field and direction of the longest axis was 45 degrees. Its shape can be seen in Figure 1. There are multiple non-uniform LEM states for the zero-field scenario (Figure 1a–c), for example: two-domain, single-vortex, and double-vortex configurations, which are all non-uniform. These were determined by running the model many times from various randomized starting states. Larger micron-sized magnetite particles will also be magnetically non-uniform (multi-domain) when in zero-field conditions, as increasing size increases non-uniformity. However, applying a high enough strength magnetic field has the effect of saturating the particles as is shown (Figure 1d–i). We did this by taking the two-domain solution and applying an increasing magnetic field strength to see the effect on the magnetic texture of the particle. It was found that the applied field effectively pushes the domain walls out of the grain (de-nucleation) gradually; at 100 mT the domain structure became approximately uniform. At 300 mT the magnetic structure saturates, and the magnetization direction aligns with the field. This was observed for other initial remanence states.

Next, we consider the magnetic field morphology of a large magnetically saturated irregular structure. Models of regular shapes, e.g., cubic, octahedral, are commonly considered in the literature [19], however, it is irregular shapes that we commonly find in mineral processing, e.g., mineral particles after crushing and grinding. We simulate this structure in a field of 100 mT, based on the micromagnetic modelling. The micromagnetic modelling (Figure 1) was for a much smaller particle; larger particles are likely to saturate in even smaller fields. We do not consider a field of 300 mT, because this is energetically more expensive, and we require that inter-grain interaction field gradients are relatively large. We modeled the magnetization of a typically shaped magnetite particle, analyzed the magnetic field of this magnetite particle, and compared it to that of a perfect magnetite sphere of the same size (Figure 2).

The geometrical features of a 'typical' particle (Figure 2a) in mineral processing are as follows: length 179.5 μm, width 113 μm, height 103 μm. The calculation parameters for the model are: (1) the geometry center of particle is at the origin of the spherical coordinate system, (2) the volume radius (r) of particles is 62 μm, (3) the external magnetic flux density is 100 mT, the average susceptibility of magnetite is $967 \times 10^{-6}$ m$^3\cdot$kg$^{-1}$ [20], (4) the isosurface of zero-magnetic potential is the spherical surface with the radius 500 μm to the origin, and (5) the average element quality is 0.8074 with the longest edge being 2 μm.

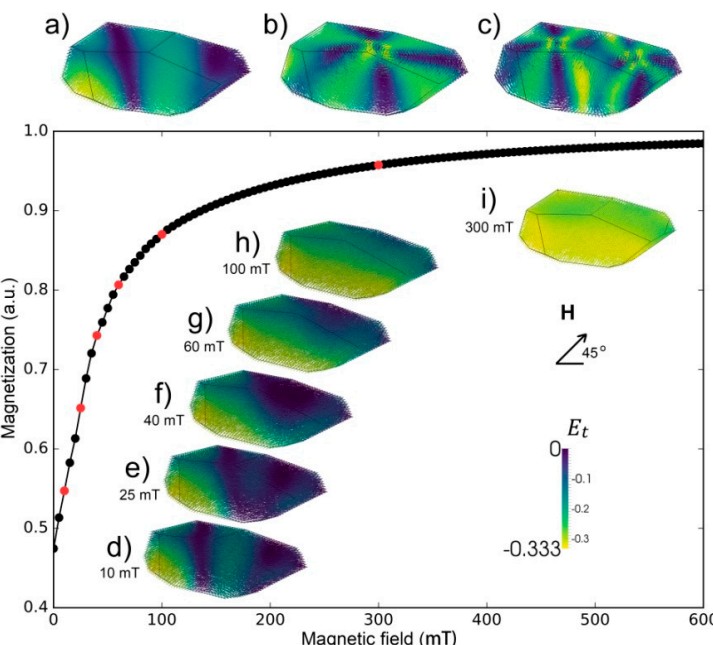

**Figure 1.** Micromagnetic model of the magnetization and magnetic structure of a magnetite particle in zero field and a field. 'Ek' is the magnetocrystalline anisotropy energy. (**a**–**c**) three zero-field structures determined from different initial random starting states: (**a**) 'two-domain' state, (**b**) single-vortex state, and (**c**) double-vortex state. (**d**–**i**) magnetic structures along the applied-field magnetization curve for the 'two-domain' state. The red dots correspond to the images (**d**–**f**) etc. The magnetic structures are colored by the local magnetocrystalline anisotropy field energy; dark blue areas are 'domain walls', and yellow areas 'domains'. The magnetization is normalized by the particles' saturation magnetization.

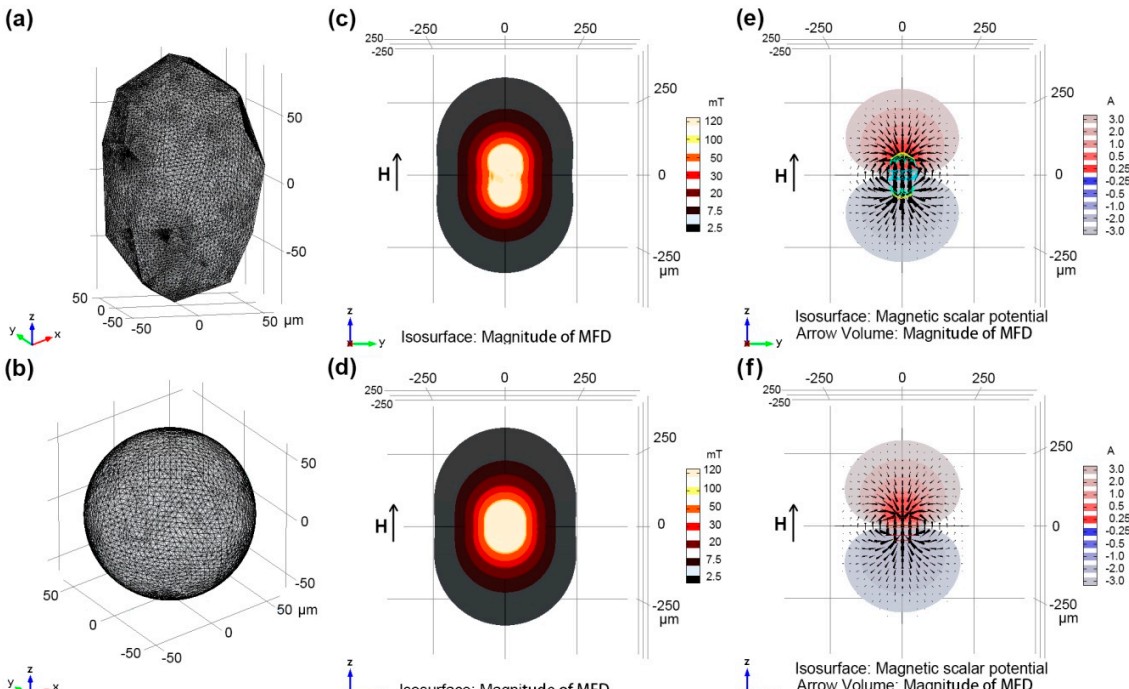

**Figure 2.** COMSOL modeling and simulation results of the magnetite particle and its equal volume sphere. (**a**,**b**) The geometric entities after mesh division; (**c**,**d**) the magnitude of magnetic flux density (MFD) in z, y plane; (**e**,**f**) the corresponding magnetic scalar potential. The applied magnetic field direction (**H**) is depicted.

As can be seen from Figure 2, the general characteristics of the particle MFD and corresponding equal volume sphere are similar. In the 3D space outside of the particles, the magnitudes of MFD are almost the same when being distal to the particles. The two particles also have similar magnetic scalar potential morphologies. Therefore, we conclude visually that the further the distance from the magnetic particles, the greater the similarity between the irregular and spherical particles' external magnetic fields.

Considering the uncertainty in the direction and location of particles in flow fields, particles may touch. Therefore, it is important to analyze the MFD of particles over a range of distances. For a comprehensive understanding of the MFD distribution of particles in 3D space, we calculated the magnitudes of MFD from the point of intersection of a coordinate axis and the surface of a particle to the point of intersection of a coordinate axis and the spherical surface with a radius of 500 μm (Figure 3a–c).

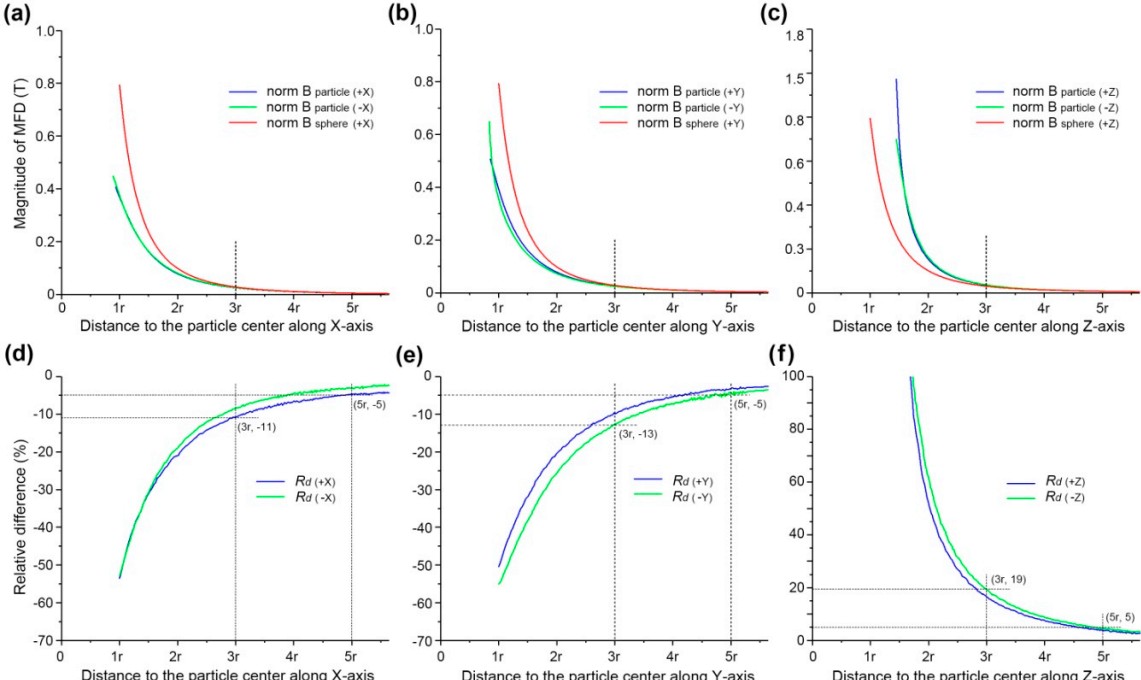

**Figure 3.** MFD characteristics of an irregular magnetic particle. (**a**–**c**) magnitude of MFD of magnetite particle along X, Y, and Z axes, respectively; (**d**–**f**) relative difference between magnitude of particle MFD and sphere MFD along X, Y, Z axes, respectively. The 'r' used in the figures' horizontal axis is the particle volume radius, and the abscissa initial value of any curves is for the intersection point of coordinate axis and particle surface.

As can be seen from Figure 3a–c, the magnitudes of MFD of their regular particle and sphere drop rapidly with the increase in distance before reaching a plateau gradually. Additionally, the magnitude of MFD along the +X and −X axis differs significantly because of the particle asymmetry when the distance is relatively small. However, when the distance is >3r, magnitudes of MFD of the particle and sphere are nearly equal.

Figure 3d–f clearly show that the relative difference between particle MFD and sphere MFD varies with the increase in distance from the particle center along the ±X, ±Y, and ±Z axes. The relative difference between particles MFD along the +X-axis and −X-axis has the same trend (Figure 3d). Furthermore, the MFD relative difference along the X-axis and Y-axis is negative while the relative difference of the particle MFD along the Z-axis is positive, because the magnitude of MFD in the vicinity close to the bulge or protrusion is greater than that of other places based on the magnetic field energy distribution difference in 3D space (Figure 2a).

The absolute relative difference along the "±"X, "±"Y, and "±"Z axes decreases dramatically when the distance is <3r, above this the difference is less defined, i.e., the absolute value of the relative difference along the three axes drops from about 10~20% to less than 5% when the distance increases from 3r to 5r. As first approximation, for non-touching particles the sphere MFD is preferred in the modeling and simulation compared with the asymmetric particle MFD for the sake of simplification.

## 4. Conclusions

The magnetization of a magnetite particle was modeled and the magnetic field of an irregular particle analyzed using FEM. The results presented in this paper indicate: the magnetization in an external applied field makes the magnetic field of the particle similar to the magnetic field of a magnetized sphere with the same volume.

(1) Two-domain, single- or double-vortex configurations were found in the magnetite particle (297.8 nm) without an external field. The non-uniformity gradually changed with increasing applied fields; at 100 mT the magnetic structure becomes approximately uniform. For larger particles, fields of <100 mT are likely required to saturate the magnetic structures.

(2) Magnitude of MFD of a particle and the corresponding sphere drop rapidly with increasing distance before reaching a plateau. The magnitude of MFD differs significantly between the particle and sphere when the distance is relatively small, e.g., close to the surface of the particle; however, when the distance is >3r, the magnitude of MFD of the particle and sphere are nearly equal.

(3) Relative differences between particle and sphere MFD vary significantly with an increase in distance from the particles.

(4) The ideas developed in this paper have the potential to improve magnetic mineral extraction yield, critical to mineral processing.

**Author Contributions:** Conceptualization and methodology, J.K.; Contributions to the micromagnetic calculation, M.A.V.-G.; Magnetic field simulation, R.D.; Writing—Original Draft Preparation, Q.C. and H.L.; Writing—Review & Editing, A.R.M. and W.Z.

**Funding:** This research was funded by [the National Natural Science Foundation of China] grant number [51674091; 51104048], [the Natural Science Foundation of Fujian Province of China] grant number [2017J01483] and [the National Training Program of Innovation and Entrepreneurship for Undergraduates] grant number [201810386029].

**Acknowledgments:** The Authors would like to acknowledge Ph. D. Adrian R. Muxworthy from Imperial College London, who provided a methodology for investigating.

**Conflicts of Interest:** The authors declare no conflict of interest.

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
