# Peer review of "Modelling External Magnetic Fields of Magnetite Particles: From Micro- to Macro-Scale"

_geosciences, doi:10.3390/geosciences9030133_

Round 1
Reviewer 1 Report
The paper is difficult for a reader because it appears to be composed of parts written by several people. For instance, the reference to [O Conbhui] is given in three versions (all inaccurate).
The eq. (2) is not sufficiently explained, contains terms of different dimensions and undefined symbols.
The terminology used for the magnetic field B (Fig 1) and the abbreviation MFD used later is not explained.
What is the mathematical basis for the COMSOL solution?
How is the quantity Ek in Fig. 1 computed?
The magnetization in Fig. 1 is normalized by what quantity?
The mass susceptibility for magnetite used (p.4) appears to be about 2 orders of magnitude lower than given in other sources.
The numerical solution for a 'perfect magnetite sphere' could be given as an analytical expression
available, e,g., in Smythe W.R. Static and dynamic electricity (1950).
Author Response
See the word document

Reviewer 2 Report
This manuscript based on mathematical and numerical modeling of magnetic particles allows to study rigorously the effects of magnetic fields applied to particles with various shapes and interactions. It is thus very usefull to the paleomagnetic community. I found the paper easy to read and well presented. I have no specific comments.
Author Response
We thank the reviewer for their comments. We make no changes in response to this reviewer.
Adrian